# Low Intra-Abdominal Pressure with Complete Neuromuscular Blockage Reduces Post-Operative Complications in Major Laparoscopic Urologic Surgery: A before–after Study

**DOI:** 10.3390/jcm11237201

**Published:** 2022-12-04

**Authors:** Claudia Brusasco, Federico Germinale, Federico Dotta, Andrea Benelli, Giovanni Guano, Fabio Campodonico, Marco Ennas, Antonia Di Domenico, Gregorio Santori, Carlo Introini, Francesco Corradi

**Affiliations:** 1Anaesthesia and Intensive Care Unit, E.O. Ospedali Galliera, 16128 Genoa, Italy; 2Urology Unit, E.O. Ospedali Galliera, 16128 Genoa, Italy; 3Department of Surgical and Diagnostic Integrated Sciences (DISC), University of Genoa, 16121 Genoa, Italy; 4Department of Surgical, Medical, Molecular Pathology and Critical Care Medicine, University of Pisa, 56126 Pisa, Italy

**Keywords:** intra-abdominal pressure, pneumoperitoneum, moderate neuromuscular blockage, complete neuromuscular blockage, post-operative acute kidney injury, post-operative complications, ERAS programs, ERAS protocols, enhanced recovery after surgery, urologic surgery

## Abstract

Most urological interventions are now performed with minimally invasive surgery techniques such as laparoscopic surgery. Combining ERAS protocols with minimally invasive surgery techniques may be the best option to reduce hospital length-of-stay and post-operative complications. We designed this study to test the hypothesis that using low intra-abdominal pressures (IAP) during laparoscopy may reduce post-operative complications, especially those related to reduced intra-operative splanchnic perfusion or increased splanchnic congestion. We applied a complete neuromuscular blockade (NMB) to maintain an optimal space and surgical view. We compared 115 patients treated with standard IAP and moderate NMB with 148 patients treated with low IAP and complete NMB undergoing major urologic surgery. Low IAP in combination with complete NMB was associated with fewer total post-operative complications than standard IAP with moderate NMB (22.3% vs. 41.2%, *p* < 0.001), with a reduction in all medical post-operative complications (17 vs. 34, *p* < 0.001). The post-operative complications mostly reduced were acute kidney injury (15.5% vs. 30.4%, *p* = 0.004), anemia (6.8% vs. 16.5%, *p* = 0.049) and reoperation (2% vs. 7.8%, *p* = 0.035). The intra-operative management of laparoscopic interventions for major urologic surgeries with low IAP and complete NMB is feasible without hindering surgical conditions and might reduce most medical post-operative complications.

## 1. Introduction

Robotic and laparoscopic procedures are currently the preferred ones for most of the major urologic surgeries because they cause less pain and require a shorter hospital stay than open surgery [1,2,3,4]. Minimally invasive surgery was first proposed by the Enhanced Recovery After Surgery (ERAS) program in 2001 for colorectal surgery and has also become a standard of care for urologic surgery in the last 5 years [5,6,7]. The ERAS protocols are continuously evolving to achieve as many benefits as possible for the patient in terms of reduced complications and a quick return to normal functions.

Generally, the level of intra-abdominal pressure (IAP) during laparoscopy has not been considered in ERAS protocols, though it may have a significant impact on surgical outcomes [8]. It is still common practice to perform surgeries with a standard IAP of 12–15 mmHg, but there is evidence to suggest that this may affect pain and reduce kidney and liver perfusions, thus being potentially detrimental to the patient [9]. A low IAP has been defined as 6–10 mmHg, and its safety and benefits have been established over a range of laparoscopic procedures, thus representing a potential complement to the ERAS peri-operative care [10]. Surgeons are often reluctant to use low IAP due to surgical space, view reduction and intra-operative bleeding for the reduced hemostatic effect of positive pressure [11,12]. A way of reducing IAP while preserving the surgical view is to maintain a complete neuromuscular blockade (NMB), which has been shown to improve surgical conditions during laparoscopic operations and also decrease post-operative pain [9]. To the best of our knowledge, no studies have investigated the effects of combining low IAP with complete NMB during laparoscopic surgery on short- and medium-term complications. 

Since April 2021, we have included low IAP in addition to complete NMB in our ERAS program (Table 1) and designed this before–after study to test the hypothesis that this strategy may result in fewer complications than a standard IAP with moderate NMB in patients undergoing major urologic surgeries by laparoscopy. 

## 2. Material and Methods

### 2.1. Study Design

This was a before–after prospective observational study conducted at the Galliera Hospital of Genova between June 2019 and December 2021. All patients signed informed consent forms on personal data storage, and the local ethics committee approved the study (7/2019 id: 4378). 

Two hundred and sixty-three consecutive patients aged > 18 yr undergoing any type of elective major urologic surgery by laparoscopy were included in the study, with no exclusion criteria (Table 2). Those undergoing surgery between June 2020 and March 2021 (before group, n = 115) were treated perioperatively with standard IAP and moderate NMB, and those undergoing surgery from April to December 2021 (after group, n = 148) were treated with low IAP and complete NMB. In the latter, low IAP was combined with the use of a novel laparoscopic CO_2_ insufflation system that improves the visual space by constantly evacuating smoke and providing a more stable pneumoperitoneum (AirSeal Intelligent Flow System^®^ or LexionSystem^®^).

### 2.2. Intraoperative Procedure

Anesthesia was induced with 1–2 mg·kg^−1^ propofol 1% and 0.2 mg·kg^−1^·min^−1^ remifentanil. The train-of-four (TOF) ratio was calibrated before the administration of NMB agents. Rocuronium 1 mg·kg^−1^ was administered, and the patients were intubated at a TOF ratio of 0. Sevoflurane (MAC 0.5-07) and remifentanil were used to maintain anesthesia, with sevoflurane MAC adjusted to target vales of the bispectral (BIS) index between 50 and 60. The TOF ratio was determined every 20 s throughout surgery and before the application of the ERAS protocol, with 0.15 mg·kg^−1^ rocuronium added if the TOF ratio was >0. After the application of ERAS protocols, a post-tetanic-count (PTC) was determined every 10 min, and 0.15–0.25 mg·kg^−1^ rocuronium was added to maintain PTC < 4, thus ensuring complete NMB. At the end of the pneumoperitoneum, no more rocuronium was administered, and soon after fascia closure, a PTC was performed, followed by an adequate dose of sugammadex to extubate the patient at a TOF ratio > 90%. In the before group, the IAP was initially set at 12 mmHg and increased to 15 mmHg, if needed; in the after group, the IAP was initially set at 7–8 mmHg and increased to 10 mmHg, if needed. In both groups, the level of IAP pressure was primarily aimed at obtaining optimal conditions in terms of view and surgical space. The AirSeal Intelligent Flow System^®^ or LexionSystem^®^ were routinely used from April 2021 during all laparoscopic surgeries. 

### 2.3. Outcome Measures

The primary outcomes were post-operative complications and hospital length-of-stay. Postoperative complications were divided into medical and surgical, depending on their nature. The Clavien–Dindo complication score was used to determine their type and severity. Secondary outcomes were operating time, estimated blood loss, respiratory dynamics changes during pneumoperitoneum (airway peak pressure, compliance, end-tidal CO_2_ (ETCO_2_)), hemodynamic intraoperative balance (mean arterial pressure and bradycardia) and total fluid administration. Postoperative anaemia was defined as hemoglobin < 10.0 g/dL or a decrease > 2.0 g/dL in the preoperative hemoglobin value.

### 2.4. Statistical Analysis

At least 175 patients were required to obtain a statistical power (1 − β) = 0.8 by assuming a proportion of total post-operative complications in the before group = 0.4 and in the after group = 0.2, with α = 0.05 and an allocation ratio = 0.8 (before group: at least 97 patients; after group: at least 78 patients) for a two-sided exact test. Categorical data are presented as the number and percentage, and continuous data are presented as the median and interquartile ranges (IQR). Categorical data were compared by Pearson’s *χ*^2^ test with Yates correction or Fisher’s exact test, when appropriate. Continuous variables were compared with the Student *t*-test for unpaired data. *p* < 0.05 was considered to indicate statistical significance. Statistical analyses were performed using the SPSS, version 27.0 (SPSS, Chicago, IL, USA) software packages.

## 3. Results

Patients’ anthropometric characteristics, type of surgery and pre-operative data did not differ significantly between groups (Table 2). All intra-operative data were similar between groups, except for the volume of fluids administered (*p* < 0.001). There were no differences in the time of surgery, respiratory and hemodynamic parameters, ETCO_2_ and intra-operative blood loss between groups (Table 3).

The total number of post-operative complications and the Clavien–Dindo severity grades were significantly lower in the low-IAP-with-complete-NMB group than those in the standard-IAP-with-moderate-NMB group (Table 4). The low-IAP group had fewer medical complications (*p* < 0.001), while surgical complications did not differ between groups (*p* = 0.422). Considering complication types, there was a statistically significant difference between groups for acute kidney injury (*p* = 0.004), persistent kidney injury at discharge (*p* = 0.045), post-operative anemia (*p* = 0.045), reoperation (*p* = 0.035) and creatinine peak during hospitalization (*p* = 0.005). According to the KDIGO classification of acute kidney injury [13], low IAP with complete NMB reduced significantly mild acute kidney injuries (KDIGO 1, *p* = 0.003). There were no significant differences in complications such as paralytic ileus, urinary fistula or hematoma. 

## 4. Discussion

The main finding of the present study in patients undergoing major laparoscopic urologic surgeries was that the use of low IAP in combination with complete NMB was associated with fewer medical complications than the standard IAP with moderate NMB. 

Nowadays, laparoscopy is the standard of care for major abdominal surgeries, as it has several advantages over laparotomy, namely, better cosmetic results, shorter hospital stays, lower postoperative pain and faster recovery. Pneumoperitoneum is essential for laparoscopic surgery, as it enables views and movements within the working site, but it has mechanical and biochemical effects that modify abdominal homeostasis.[14] The inflation pressure of 12–15 mmHg, generally used for laparoscopic surgery, corresponds to the definition of stages I-II of intra-abdominal hypertension in critical care medicine [15,16]. These levels of IAP have been shown to increase pain and decrease organ perfusion, potentially contributing to an increase in post-operative complications [17,18]. Reducing IAP by itself can reduce surgical space and may not help in controlling intra-operative blood loss; thus, strategies enabling the reduction in IAP while maintaining an adequate surgical space are required [19,20,21,22,23].

In a recent meta-analysis focusing on deep vs. moderate NMB and low vs. standard IAP, Wie et al. [9] concluded that a low IAP with a deep NMB is not significantly more effective than a standard IAP with various NMB combinations for optimizing surgical space conditions, the duration of surgery or postoperative pain, while a low IAP with a deep NMB may result in fewer peri-operative complications. The results of our present study demonstrate an overall reduction in post-operative complications in patients treated with low IAP and complete NMB throughout laparoscopic surgery, which was apparently due to a reduction in medical complications and, specifically, acute post-operative kidney injury. Pneumoperitoneum has been shown to be associated with a risk of oliguria for an IAP up to 15 mmHg or anuria at an IAP of 30 mmHg [24], possibly due to the renal blood flow being reduced by about 40%, with decreased creatinine clearance [25]. Both the level of IAP and the duration of insufflation can impact the glomerular filtration rate and urinary sodium excretion, even in cases with preserved mean arterial pressure [25]. In an animal model, Schäfer et al. found a remarkable reduction in renal filtration pressure when IAP was raised to 14 mmHg [26]. This may be due not only to the compression of the central venous system but also to that of renal parenchyma, with an impairment of renal venous drainage, lymphatic drainage or both. Although the threshold of IAP for kidney damage seems to be about 15 mmHg in humans [13,27], the presence of comorbidities, the increasing complexity of surgical interventions and prior episodes of post-operative acute kidney (PO-AKI) can make some patients susceptible to kidney damage, even at an IAP < 15 mmHg [27]. Therefore, any means of reducing IAP without hindering surgical conditions has the potential of reducing the risk of post-operative kidney injury. 

In the present study, the reduction in PO-AKI was mainly in patients at KDIGO’s stage 1, which may be the reason why the length-of-stay and Clavien–Dindo classification were not statistically different. The PO-AKI stage 1 of KDIGO classification [13] is a moderate PO-AKI that is often reversible and therefore little considered in clinical practice and often not recognized post-operatively. Unfortunately, patients with some grade of PO-AKI show significantly inferior survival rates compared to patients with normal kidney function, and recurrent AKI represents a risk factor for the future decline of kidney function [27]. Therefore PO-AKI KDIGO 1 should not be underestimated and should be recognized early in order to increase the level of patient surveillance and avoid worsening to other more severe stages of PO-AKI. The KDIGO 2–3 patients were similar in the two groups, and this is due to the quote of obstructive PO-AKI that is typical of major urological surgery, classically characterized by a rapid increase in creatinine and resolution with reoperation (Clavien Dindo 3). 

Low IAP had no effects between the two groups on intra-operative data such as respiratory mechanics, time of surgery and intra-operative blood loss. Additionally, hemodynamics did not change, with few episodes of bradycardia in both groups and the same mean arterial pressures and cardiac frequency, while a statistically different amount of intra-operative fluids was infused between groups. Fluid administration is usually the first attempt to restore the systemic hemodynamics in clinical practice, and this is carried out to avoid intra-operative prolonged hypotension, which is well known to be an independent risk factor for postoperative complications such as myocardial ischemia, AKI or mortality. Hypotension is an independent risk factor for PO-AKI through renal hypo-perfusion and a subsequent reduction in medullary blood flow. Walsh et al. [28] described how many patients with PO-AKI have had an episode of peri-operative hemodynamic instability at some stage. The AKI risk increased with the duration of MAP < 55 mmHg, with a period of hypotension of over 20 min having a relative risk of 1.51 (95%CI 1.24–1.84) [29,30]. In addition, prolonged periods of MAP ≤ 65 mmHg may increase the postoperative AKI risk. The association between AKI and intra-operative hypotension appears to be a consistent finding [29,30]. However, in our study, there was great attention devoted to avoiding hypotensive events, this being considered a goal standard in our ERAS protocol; therefore, in both groups, the MAP was never less than 65 mmHg. However, to maintain an adequate MAP in the standard-IAP group, there has been a greater infusion of liquids that might have led to fluid overload. The fluid overload and the intra-abdominal pressure from pneumoperitoneum both work against renal function, causing renal venous congestion and a consequent decline in urine output [24,27]. A restrictive intra-operative fluid administration combined with an individualized goal-directed fluid therapy following cardiac output and stroke volume in those patients at a high risk of complications should be the correct intra-operative management to avoid volume overload, as this is associated with both increased AKI and mortality, while more appropriately timed fluid is associated with reduced AKI and the use of inotropic agents [31].

No differences in intra-operative blood loss have been found between groups, while post-operative anemization and the need for re-intervention were higher in the standard-IAP-with-moderate-NMB group. Intra-operative fluid overload and the development of PO-AKI, even in the case of KDIGO 1, led to an increase in coagulopathies, anastomotic leakages and wall infections [32,33,34]. Moreover, the hemostatic effect of a higher IAP during surgery might prevent the recognition of venous bleedings, hindering a prompt mechanical or diathermic hemostasis. 

Low IAP can be applied only if an optimal surgical view and space are maintained. To this aim, we used it in combination with complete NBM and novel laparoscopic CO_2_ insufflation systems constantly evacuating smoke and providing a stable pneumoperitoneum, namely, the AirSeal Intelligent Flow System^®^ or LexionSystem^®^ [35,36]. The only functional difference between these two systems is that the latter LexionSystem^®^ adds humidification to CO_2_ [37].

The main strength of this study is a sample size much larger than those in previous studies on this topic, while the main weakness is the single-center, observational, before–after design. Moreover, although the study included all consecutive patients undergoing major urologic surgery, the number of severe complications was low. This might have been the result of an advanced ERAS protocol but limited the statistical power for detecting rare secondary outcomes. Finally, there were no obese patients in our study population, the BMI median being in normal range; therefore, the feasibility of low IAP with complete NMB in maintaining an optimal surgical view should be evaluated in future studies. Further randomized trials aimed at assessing post-operative complication rates in low IAP with complete NMB compared to standard IAP with moderate NMB are necessary. 

In conclusion, the results of this study suggest that the intra-operative management of laparoscopic interventions for major urologic surgeries with low IAP and complete NMB is feasible without hindering surgical conditions and reduces post-operative complications, particularly acute kidney injury. 

## Figures and Tables

**Table 1 jcm-11-07201-t001:** ERAS protocol.

Preoperative period	Patient counseling, education, and pre-habilitation.
	Avoid mechanical bowel preparation.
	Avoid fasting.
	Carbohydrate loading (drink 800 mL the night before surgery and 400 mL 2 h before surgery).
	Compression stockings and intermittent pneumatic compression devices.
Intra-operative period	Antibiotic prophylaxis and skin preparation.
	Blended opioid-sparing anesthesia technique with regional or plexus techniques
	Prevention of intra-operative hypothermia.
	Individualized goal-directed fluid therapy for ASA > 2 patients and restrictive fluid administration for ASA < 2 patients.
	BIS™ monitoring.
	Moderate NMB with TOF monitoring and standard IAP (until March 2021). Complete NMB with TOF/PTC monitoring and low IAP (from April 2021). Complete reversal of NMB before extubating.
Postoperative period	Avoid postoperative nasogastric intubation.
	Early oral intake.
	Early mobilization.
	Multimodal opioid-sparing analgesia combined with regional or local anesthesia.

ASA, American Society of Anesthesiology score; BIS™, Bispectral Index™ Monitoring System; NMB, neuromuscular blockage; TOF, train-of-four; PTC, post-tetanic count; IAP, intra-abdominal pressure.

**Table 2 jcm-11-07201-t002:** Baseline characteristics of patients and pre-operative data.

KERRYPNX	Standard IAP	Low IAP	*p* Value
	Moderate NMB	Complete NMB	
	(N. 115)	(N. 148)	
Sex, m/f n	95/20	124/24	0.868
Age, yr	66 (60–72)	68 (60–72)	0.404
BMI, kg/m^2^	25 (23–28)	25 (23–28)	0.578
Smoking, n	21	24	0.742
Comorbidities, n			
Hypertension	54	59	0.295
Cardiovascular disease	11	21	0.338
Respiratory disease	5	11	0.435
ECOG performance status, score	0 (0–0)	0 (0–0)	0.577
CCI, score	4 (3–5)	4 (2–5)	0.873
CCI estimated 10 yr survival, %	53 (21–77)	53 (21–90)	0.558
ASA physical status, class	2 (2–2)	2 (2–2)	0.880
Hemoglobin, g·dL^−1^	14.4 (13.3–15.2)	14.5 (13.4–15.3)	0.953
Creatinine, mg·dL^−1^	1.0 (0.8–1.1)	0.9 (0.8–1.0)	0.124
Type of surgery, n (%)			
Partial nephrectomy	26 (22.6%)	27 (18.2%)	
Radical prostatectomy	60 (52.2%)	88 (59.4%)	
Radical nephrectomy	20 (17.4%)	21 (14.3%)	
Adrenalectomy	4 (3.5%)	1 (0.7%)	
Radical cystectomy	1 (0.8%)	1 (0.7%)	
Other	4 (3.5%)	10 (6.7%)	

BMI, body mass index; ECOG, Eastern Cooperative Oncology Group; CCI, Charlson comorbidity index; ASA, American Society of Anesthesiology. Data are the median with the interquartile range (IQR) or absolute numbers with the percentage (%).

**Table 3 jcm-11-07201-t003:** Intra-operative data.

	Standard IAP Moderate NMB (N. 115)	Low IAP Complete NMB (N. 148)	*p* Value
Duration of surgery, min	130 (87–160)	130 (102–165)	0.868
Duration of pneumoperitoneum, min	100 (57–130)	104 (70–135)	0.725
Duration of anesthesia, min	165 (125–195)	167 (135–200)	0.553
Blood loss, ml	200 (90–400)	200 (50–350)	0.900
Peak airway pressure, cmH_2_O			
before pneumoperitoneum	15 (13–16)	14 (13–16)	0.814
during pneumoperitoneum	20 (18–22)	19 (17–22)	0.056
Static respiratory compliance, mL·cmH_2_O^−1^			
before pneumoperitoneum	61 (52–76)	66 (55–75)	0.109
during pneumoperitoneum	37 (31–43)	38 (32–45)	0.412
ETCO_2_, mmHg	34 (32–35)	34 (32–36)	0.226
Oxygen saturation, %	99 (98–99)	99 (98–99)	0.845
Heart rate, min^−1^	60 (54–65)	60 (54–65)	0.355
Mean arterial pressure, mmHg	76 (70–85)	78 (72–85)	0.568
Intra-operative fluids, L	1.50 (1.20–2.00)	1.30 (1.00–1.58)	<0.001
Bradycardia, n	12	13	0.998
Subcutaneous emphysema, n	1	1	0.999

Data are the median with the interquartile range (IQR) or absolute numbers with the percentage (%).

**Table 4 jcm-11-07201-t004:** Outcomes and post-operative data.

	Standard IAP Moderate NMB (N. 115)	Low IAP Complete NMB (N. 148)	*p* Value
Total post-operative complications, n (%)	48 (41.7%)	33(22.3%)	<0.001
Medical complications, n (%)	34 (29.6%)	17 (11.5%)	<0.001
Surgical complications, n (%)	12 (10.4%)	14 (9.5%)	0.837
Clavien–Dindo class, n (%)			
0	65 (56.5%)	108 (73.0%)	0.006
1	26 (21.4%)	21(14.2%)	0.104
2	15 (15.9%)	16 (10.8%)	0.700
3 (A-B)	9 (7.1%)	3 (2%)	0.035
ICU admittance, n	1 (0.9%)	1 (0.7%)	0.998
Paralytic ileus, n (%)	3 (2.6%)	2 (1.3%)	0.657
Urinary fistula, n (%)	4 (3.5%)	4 (2.7%)	0.774
Hematoma, n (%)	4 (3.5%)	2 (1.3%)	0.410
Acute kidney injury, n (%)	35 (30.4%)	23 (15.5%)	0.004
KDIGO grades, n (%)			
1	27(23.5%)	14 (9.5%)	0.003
2	3 (2.6%)	6 (4.1%)	0.735
3	5 (4.3%)	3 (2%)	0.303
Persistent kidney injury at discharge, n (%)	20	14 (9.5%)	0.065
Postoperative anemia, n (%)	19 (16.5%)	10 (6.8%)	0.049
Reoperation, n (%)	9 (7.8%)	3 (2%)	0.035
Peak creatinine, mg·dL^−1^	1.1 (0.9–1.4)	1.0 (0.9–1.2)	0.005
Creatinine at discharge, mg·dL^−1^	1.0 (0.8–1.2)	0.9 (0.9–1.1)	0.045
Hospital length-of-stay, days	5 (4–7)	4 (3–5)	0.143

ICU, Intensive care unit; KDIGO, Kidney Disease: Improving Global Outcomes. Data are the median with the interquartile range (IQR) or absolute numbers with the percentage (%).

## Data Availability

The data were not inserted in publicly archived datasets but are available in an anonymized form for research purposes on request to the corresponding author.

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
