# Peer review of "Low Intra-Abdominal Pressure with Complete Neuromuscular Blockage Reduces Post-Operative Complications in Major Laparoscopic Urologic Surgery: A before–after Study"

_jcm, 2022, doi:10.3390/jcm11237201_

Round 1

Reviewer 1 Report

This is a topical question that has been asked and studied.

My only major concern with this study is that the methods of data collection used to observe and record the post operative outcomes are not clearly stated. I assume it was from retrospective review of the medical records after 30 days from operation but this is not stated. 

The definition of anaemia used in this study should be stated.

The paper is well written and the conclusions are logically discussed.

Author Response

We thank the reviewer for his suggestions. The study was a prospective observational study with a before-after analysis. This has been added at the beginning of the methods section (page 3 line 72)

Postoperative anaemia has been defined in the “outcome measures” paragraph. Anaemia was defined as a postoperative hemoglobin less than 10,0 g/dL or a loose from basal hemoglobin of 2,0 g/dL (for example if patient had a basal of 100 g/L and postoperatively hemoglobin was 9 g/L no anaemia was recorded.) page 4 line 112. 

Reviewer 2 Report

well written, good and scientifically sound discussion. My only comment is that both groups do not include patients with high BMI's (BMI is 25 in both groups). So this could be a contributing factor in having a good peroperative vision

Author Response

We thank the reviewer for his comments. Yes we agree completely that these results and mostly fesability on obese patients  should be serched and described. This issue has been added in the limitations section of the discussion and could be a good idea for further studies. (Page 7 lines 237-240)